# Lung Cancer Mortality Trends in a Brazilian City with a Long History of Asbestos Consumption

**DOI:** 10.3390/ijerph16142548

**Published:** 2019-07-17

**Authors:** Gisele Aparecida Fernandes, Eduardo Algranti, Gleice Margarete de Souza Conceição, Victor Wünsch Filho, Tatiana Natasha Toporcov

**Affiliations:** 1Department Epidemiology, School of Public Health, University of São Paulo, São Paulo-SP 01246-904, Brazil; 2Division of Medicine, Fundação Jorge Duprat e Figueiredo (Fundacentro), São Paulo-SP 05409-002, Brazil

**Keywords:** asbestos, lung neoplasms, mortality

## Abstract

There are scarce epidemiological studies on lung cancer mortality in areas exposed to asbestos in developing countries. We compared the rates and trends in mortality from lung cancer between 1980 and 2016 in a municipality that made extensive use of asbestos, Osasco, with rates from a referent municipality with lower asbestos exposure and with the rates for the State of São Paulo. We retrieved death records for cases of lung cancer (ICD-9 C162) (ICD-10 C33 C34) from 1980 to 2016 in adults aged 60 years and older. The join point regression and age-period-cohort models were fitted to the data. Among men, there was an increasing trend in lung cancer mortality in Osasco of 0.7% (CI: 0.1; 1.3) in contrast to a mean annual decrease for Sorocaba of -1.5% (CI: −2.4; −0.6) and a stable average trend for São Paulo of -0.1 (IC: −0.3; 0.1). Similar increasing trends were seen in women. The age-period-cohort model showed an increase in the risk of death from 1996 in Osasco and a reduction for Sorocaba and São Paulo State during the same period. Our results point to a need for a special monitoring regarding lung cancer incidence and mortality in areas with higher asbestos exposure.

## 1. Introduction

Lung cancer is an important cause of death, with an estimated 1.8 million deaths worldwide in 2018, 82,000 of which occurred in South America and the Caribbean [1]. In 2016, the Brazilian National Cancer Institute (INCA) notified 27,000 deaths in the country [2]. 

Lung cancer is associated with tobacco smoking, socio-economic factors, air pollution, exposure to occupational carcinogenic agents, and ionizing radiation [3,4,5]. The increase in consumption of tobacco in the last century was accompanied by a significant increase in the incidence of pulmonary neoplasms. While the effect of occupational exposures is smaller than that of tobacco, it is superior to other classes of risk associated with lung cancer [6].

Asbestos, both chrysotile and the amphiboles, is associated with increased lung cancer risk [7]. Global estimates of annual deaths due to lung cancer related to asbestos exposure vary between 90,000–191,000 [8,9]. The fraction of lung cancer incidence attributable to asbestos exposure varies between 4 and 10% in most developed countries [10]. Asbestos exposure and smoking act synergistically to increase lung cancer risk [11].

Brazil is one of the largest worldwide producers, consumers, and exporters of chrysotile asbestos. Brazil’s use of chrysotile asbestos began in the late 1930s. Since the 1970s, asbestos consumption grew intensely and peaked at 1.5 to 1.6 kg/inhabitant between 1985 and 1991. In 2015, Brazilian asbestos production was responsible for 15.4% of the world fibre production [12]. Routine quantitative fibre measurement in workplaces that began in the 1980s was essentially limited to asbestos mining and asbestos-cement industries [13,14].

Since the mid-1980s, several countries have banned the use of the fibre in their territories because of consistent epidemiological evidence of its health risks [15]. In November 2017, the Brazilian Supreme Court rendered a decision prohibiting the extraction, industrialization, and commercialization of asbestos throughout the Brazilian territory [16]. However, the decision will not be implemented until the industry appeals to the court are judged.

In Brazil, there is a strong preponderance towards the consumption of chrysotile [14]. Amphibole asbestos was used in small quantities for asbestos-cement pipes until the beginning of the 1980s [13]. Likewise, this increased chrysotile consumption resulted in a higher ratio of lung cancer to mesotheliomas compared with countries that had relevant amphibole production or consumption [17]. Nevertheless, in Brazil, scarce epidemiological studies on the relationship between asbestos exposure and lung cancer mortality have been published. 

The aim of this study is to compare the rates and trends in mortality from tracheal, bronchial, and lung cancer between 1980 and 2016 in a municipality that made extensive use of asbestos in the last 60 years, Osasco, with rates from a referent municipality, where asbestos consumption has been much less and with the rates for the State of São Paulo. 

## 2. Materials and Methods 

### 2.1. Study Design

We conducted an ecological study using information on lung cancer mortality and population estimations of two large cities in the State of São Paulo and for the whole State. Data were extracted from the Mortality Information System (SIM) of the Ministry of Health, provided by the National Health System [18].

### 2.2. Geographical Areas

Historically, the state of São Paulo has housed almost 50% of all asbestos-consuming industries and the largest asbestos-cement plant in Latin America that operated in the municipality of Osasco for more than 50 years. The municipality of Osasco is located in the State of São Paulo, with a population of 320,436 men and 346,304 women in the year 2010 [19]. Between 1941 and 1994, an estimated 10,700 employees worked in asbestos-cement plants. 

Sorocaba, located 86 kilometres from Osasco, was chosen as a referent municipality given its similar population size, similar Human Development Index (HDI), and much less asbestos consumption from one large and two small brake and clutch lining plants. The Sorocaba population included 287,014 men and 299,611 women in the year 2010 [19]. 

The state of São Paulo population was comprised with 20,077,873 men and 21,184,326 women in the year 2010 [19]. The values for the HDI for Osasco, Sorocaba and the State of São Paulo were similar throughout the period, considering the censuses from 1991 (0.572, 0.579, and 0.578), 2000 (0.706, 0.721, and 0.702) and 2010 (0.776, 0.798, and 0.783), respectively [19].

### 2.3. Data Extraction and Analysis

Mortality rates due to tracheal, bronchial and lung cancer in Osasco were compared with those for Sorocaba and the State of São Paulo.

From 1980 to 1995, the underlying cause of death was retrieved from the SIM using the International Classification of Diseases 9th revision (ICD-9 C162). Data from 1996 to 2016 were coded under the 10th revision (ICD-10 C34). As tracheal cancer was separated from lung and bronchial cancer in ICD 10, to correct this divergence, we added C33 malignant tracheal neoplasm to C34.

Population data were obtained from the electronic pages of Datasus provided by the Brazilian Institute of Geography and Statistics (IBGE) for the censuses conducted in 1980, 1991, 2000, and 2010 [20]. For intercensitary years, we used projections from the same data source. Deaths in which sex and age (0.04%) data were missing were excluded from the analysis. 

Age-standardized mortality for the age group 60 and older was calculated for each sex. Age standardization was performed using the direct method with the standard world population modified by Doll [21]. For time trend analysis, we obtained the rates for each year using a simple five-year moving average. 

We plotted the rates and chose the regression model with the better goodness-of-fit as measured by the R^2^ value. Significant trends were considered as having a p-value ≤0.05. A joinpoint [22] analysis was fitted to calculate the mean annual percentage change (AAPC) and the respective 95% confidence intervals. 

Age-period-cohort analysis used Poisson regression models. Data were grouped into five-year categories: For age- from 60–64 years to 80 years and over; for the period 1981–1985 to 2011–2015; and for the birth cohort from 1901–1905 to 1951–1955. The difference between period and age group resulted in the value for birth cohort. 

In the model used, the effects act multiplicatively, so the expected value for the rate logarithm is a linear function of the age, period and cohort effect [23,24]. The age-period-cohort model splits time into three axes: Age, period, and cohort. These models were built for verification, in particular in our study, of the period effect. Such an effect often results from external factors affecting the age groups over a specific period of time. In the cohort effect, similar health outcomes are observed in groups of individuals who were born in similar years. In the effect of age, the influence of time on health outcomes is observed.
(1)$$\left[log\;\lambda \;\left(a,p\right)=\alpha_{a}+\beta_{p}+\gamma_{c}\right]$$
where *λ* is the rate formula and *α_a_*, *β_p_*, and *γ_c_* are the functions of age (*a*), period (*p*), and birth cohort (*c*). We used the median values for cohort (1936–1940) and period (2001–2005) as reference. There was a limitation for the estimation of the complete model, due to the exact linear relationship of age, period, and cohort effects. To circumvent this limitation and to calculate the effect parameters, we used the method proposed by Holford in which the estimable functions are deviations, curvatures, and derivations (drift) [23].

We selected the model that presented the lower deviance, as a sign of better goodness-of-fit. The significance level was set at 5%. The relative risk (RR) (with 95% confidence intervals) was the association measure, with RR = 1 being the reference cohort and RR = 1 being the reference period.

The analyses for age-period-cohort effect were performed using R Statistical Software, version 3.5.2. The research ethics committee of the School of Public Health of the University of São Paulo approved this study under the protocol number 2.518.202 in 2018.

## 3. Results

During the 37 years considered (1980–2016), 967 (men), and 444 (women) deaths due to lung cancer were registered in Osasco, whereas 905 and 399 occurred in Sorocaba and 73,308 and 33,406 in the State of São Paulo. The highest age-standardized mortality rates for lung cancer in old men were in Sorocaba in the first period (1980–84) and in Osasco in the last period (2012–2016). Between 1980 and 2016, Osasco showed a mean annual increase of 0.7% (CI: 0.1; 1.3) in age-standardized death rates, in contrast to a decrease in Sorocaba of –1.5% (CI: −2.4; −0.6) and a stable trend in São Paulo of −0.1% (CI: −0.3; 0.1) (Table 1).

Figure 1 presents scatter plots with detailed modelled trend curves for men and women. The best fitting trend lines for the period 1980–2016 were a polynomial regression model for males and a simple linear curve for females. Lung cancer mortality among men increased in all locations in the beginning of the period, with a peak at the end of the 1990s followed by a subsequent decrease. This reduction was more discrete in Osasco, where the rates surpassed those in Sorocaba and São Paulo. The differences in trends according to location remained in stratified analyses by age group (five-year intervals) (Figure 2).

Among women aged ≥60 years, the highest age-standardized mortality rates for lung cancer were in São Paulo in the first period and in Osasco in the last period (Table 1). In females, similar growing trends for Osasco, Sorocaba, and São Paulo were observed, with a slightly higher increase in Osasco (Figure 1).

In the age-period-cohort analysis, the highest age-standardized mortality rates by lung cancer in men occurred in similar cohorts for all locations: For those born between 1916 and 1935 in Osasco; between 1926 and 1940 in Sorocaba; and 1921 and 1940 in São Paulo State (Figure 3) (Appendix A).

When assessing the period effect, rates due to lung cancer increased in old men from Osasco between 1996 and 2015, with the exception of a decrease in the period during 2006–2010. Conversely, the risk of death by lung cancer decreased among old men in Sorocaba, remained stable in the state of São Paulo in the periods 1996–2000/2001–2005 and decreased in 2006–2010 (Figure 3) (Appendix A). The age-period model, the drift and the age-period-cohort showed the best goodness-of-fit for Osasco (*p* = 0.076), Sorocaba (*p* < 0.001), and the State of São Paulo, respectively (*p* < 0.001) (Table 2).

Among women, the age-cohort period analyses revealed similar increases in mortality for the cohorts born after 1940 and for the periods after 2000 in all locations assessed (Appendix A). 

## 4. Discussion

This study shows an increasing trend in lung cancer mortality rates in men aged ≥60 years living in Osasco, in contrast with declining rates in Sorocaba and the state of São Paulo from 1980 to 2016. Age-period-cohort analysis disclosed an increased risk of death by lung cancer in the period from 1996 onwards in Osasco, and this trend was not replicated in Sorocaba and the state of São Paulo. Among women, lung cancer mortality increased for all locations.

The increase in lung cancer mortality in women is consistent with previous international studies [16,25] and is frequently attributed to a growing prevalence in smoking among females in the last decades. Considering that lung tumours are frequently related to exogenous risk factors [26], the agreement in trend directions across Osasco, Sorocaba, and São Paulo might reflect similarities in lifestyles and/or risk exposures in the decades before deaths occurred. The reduced number of cases in Osasco and Sorocaba may explain higher oscillations in rates in these municipalities compared to the whole State. 

The decrease in lung cancer mortality in men after the end of the 1990s in all locations agrees with results obtained in global studies [27]. The International Burden of Diseases International classifies São Paulo State as having a socio-demographic development index of “medium to medium-high”, for which reducing trends for lung cancer mortality in men have been reported [16,25,27]. A plausible explanation for this finding is the decreasing prevalence in tobacco consumption in Brazil [28]. Smoking is considered a major risk factor for lung cancer incidence. The implementation of successful public policies for tobacco control in the country led to a decline in smoking trends since the end of the last century in all Brazilian regions [28,29,30,31].

The trend analysis revealed that death rates due to lung cancer decreased less in Osasco and surpassed those in Sorocaba and São Paulo after 1996. Analysis of the period effect in old males also showed an increase in rates for almost all the periods evaluated since 1996–2000 in Osasco as opposed to the two other locations. A possible explanation for the unexpected trend in Osasco might be a higher exposure to risk factors for lung cancer in the 1950/1960s (considering exposures with a latency period of 40 years) or 1970/1980s (considering exposures with a latency period 10 or 20 years) compared to the other locations under study. Discrepancies in mortality trends for lung cancer across regions studied did not hold among women, thus pointing to the possibility that the differences in exposure to risk factors would have been more important among men. Asbestos has a latency period between 10 and 40 years for lung cancer [32,33,34] and Osasco harboured one of the largest asbestos-cement industries in the State of São Paulo. Approximately 99% of its plant workers were men, and the peak of exposure occurred in the 1980s. The estimated prevalence of asbestos exposure in the city of Osasco peaked at 1.27%, in 1988, among the economically active men. Previous studies suggested an increased number of lung cancer cases in locations exposed to chrysotile, which is carcinogenic to the lung [17,35]. Considering all this information, our results might be compatible with the hypothesis that areas with high asbestos consumption could be related to increased lung cancer mortality in the population.

Studies assessing the occurrence of lung cancer in the overall population in cities housing asbestos mining or asbestos industries are rare, given the methodological limitations and challenges for assessing the exposure level and avoiding confusion due to tobacco consumption [36]. A Dutch study using the age-period-cohort method to predict the future numbers of asbestos-related lung cancers from 2011 to 2030 in men and women found a minimal risk of the disease in people born after the 1970s, with a peak in 2022 [36]. Ecological studies assessing the trends and age-period-cohort effect on the risk of mesothelioma in a population with a higher risk of asbestos exposure are more common. In Germany, a minimal risk of mesothelioma death was found in men and women born after the mid-1940s, with a peak of deaths in 2020 [37]. In Italy the risks declined in men born after 1945, and the peak of mortality was identified between 2012 and 2024 [38]; in Spain the risk increased for men born between 1937 and 1947, with a peak of mortality in 2016 [39]. In Brazil, a study estimated that mesothelioma mortality should reach its peak between 2021 and 2026, and an excess of mesothelioma deaths was found in Osasco when compared with the whole country [14].

Ecological studies are limited for avoiding confusion, especially for complex diseases. Since lung cancer is highly associated with tobacco smoking and air pollution, the lack of data on this specific risk factor for Osasco, Sorocaba, and the State of São Paulo is a weakness of the study. Only data on smoking in São Paulo city (the State major city) is available from the 1970s, and it shows a reduction in smoking prevalence in men (54% in 1971, 32% in 1987, 25.5% in early 2001), and a decrease in women (20%, 32%, and 19.8%, in 1971, 1987, and 2001, respectively) [40,41,42]; see Appendix A. In this study, we presumed that the trends for tobacco smoking would be expected to be the same across the locations and similar to those in São Paulo City. A previous study showed dependence among lung cancer mortality trends, human development, and smoking [43,44,45]. The high similarity in the HDI of the State of São Paulo, Osasco, Sorocaba and the city of São Paulo would be a strategy to reduce confusion. With respect to air pollution, the Environmental Company of the State of São Paulo (CETESB) reported that the level of particulate matter air pollution 10 (<10 µm) decreased in the cities studied from 2000 to 2017 (Osasco = −46%; Sorocaba = −38%) [46]. Another study limitation is the lack of comparison between lung cancer and mesothelioma mortality in the areas assessed. The number of mesothelioma cases is frequently used to estimate the number of lung cancer cases, the ratios varying according to the main fibre type consumed. In Brazil, most of the asbestos used was chrysotile. We were not able to assess age-period-cohort effects of mesothelioma mortality because of the reduced number of cases in the studied areas [17].

Another inherent limitation of ecological studies refers to the impossibility of individual inferential causality. We cannot be sure whether individuals with lung cancer were those exposed to risk factors. However, this design allows for relevant population inferences so that our results point to a need for a special monitoring regarding lung cancer incidence and mortality, as well as health assistance, in areas with higher asbestos exposure.

## 5. Conclusions

There is evidence of an increased risk of death due to tracheal, bronchial and lung cancer in males, specifically in Osasco, a city that housed a large cement-asbestos industry in the last century. Our study points to the possibility of increased mortality due to lung cancer in cities exposed to asbestos, especially chrysotile. The surveillance of lung cancer cases and deaths in these areas is strongly recommended. In addition, the surveillance of the prevalence of major risk factors for lung cancer—such as tobacco smoking, asbestos exposure, ionizing radiation, and air pollution—is important to increase the accuracy for more specific analysis of the fibre effect on lung cancer.

## Figures and Tables

**Figure 1 ijerph-16-02548-f001:**
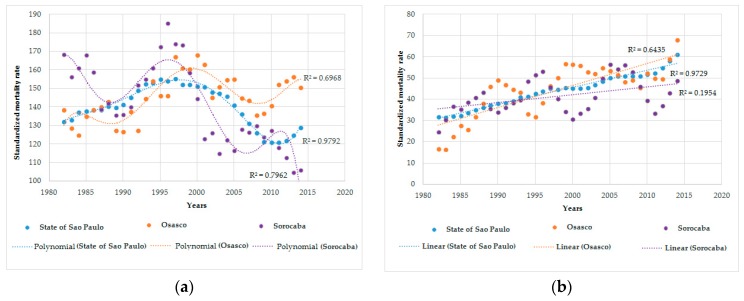
Age-standardized mortality rates for trachea, bronchi and lung cancer in people aged ≥60 years. State of São Paulo-Brazil, 1980-2016: (**a**) Standardized mortality rate for trachea, bronchial, and lung cancer in men; and (**b**) standardized mortality rate for trachea, bronchial, and lung cancer in women.

**Figure 2 ijerph-16-02548-f002:**
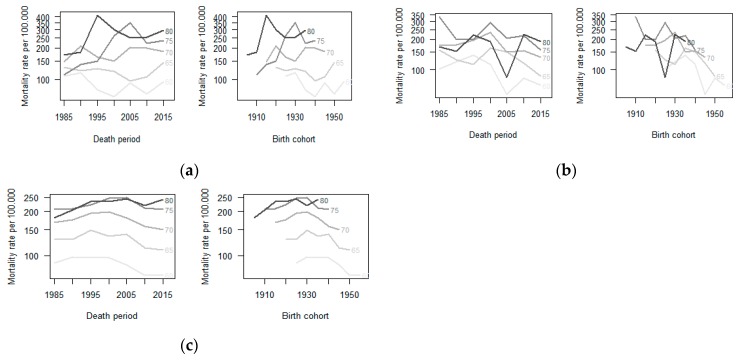
Age-specific mortality rate from trachea, bronchus and lung cancer by period and birth cohort in men aged ≥60 years. São Paulo-Brazil 1980–2016: (**a**) Age-specific mortality rate due to trachea, bronchus and lung cancer by period and birth cohort in Osasco. (**b**) Age-specific mortality rate due to trachea, bronchus, and lung cancer by period and birth cohort in Sorocaba. (**c**) Age-specific mortality rate due to trachea, bronchus, and lung cancer by period and birth cohort in the State of São Paulo-Brazil

**Figure 3 ijerph-16-02548-f003:**
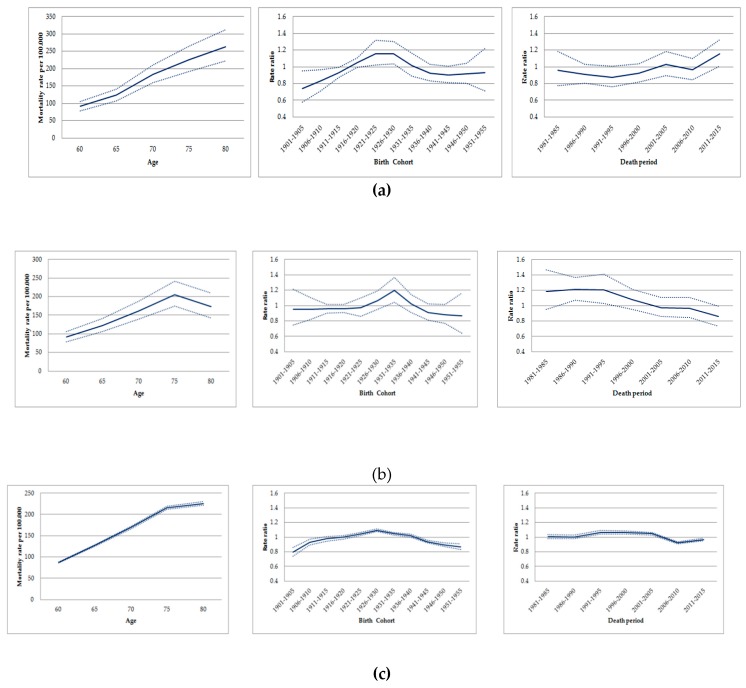
Age-cohort-period effects on mortality due to trachea, bronchial and lung cancer in men aged ≥60 years. São Paulo-Brazil, 1980-2016: (**a)** Age-cohort-period effects on mortality due to trachea, bronchial, and lung cancer in Osasco. (**b**)Age-cohort-period effects on mortality due to trachea, bronchial, and lung cancer in Sorocaba. (**c**)Age-cohort-period effects on mortality due to trachea, bronchial, and lung cancer in São Paulo State.

**Table 1 ijerph-16-02548-t001:** Trends in mortality rates due to trachea, bronchial and lung cancer, men and women aged ≥60 years. São Paulo-Brazil, 1980–2016.

Place	ASR ^1^				AAPC ^2^	
	Male		Female		Male	Female
	1980–1984	2012–2016	1980–1984	2012–2016	1980–2016	1980–2016
**Osasco**	138.47	150.54	16.72	67.80	0.7 * (0.1 to 1.3)	4.7 * (2.9 to 6.5)
**Sorocaba**	168.50	106.9	24.78	48.78	–1.5 *(–2.4 to -0.6)	1.5 (-0.9 to 3.9)
**São Paulo State**	132.09	129.01	31.75	60.88	–0.1 (–0.3 to 0.1)	2.2 *(1.9 to 2.5)

^1^ ASR: age-standardised rate; ^2^ AAPC: average annual percentage change; *: statistically significant.

**Table 2 ijerph-16-02548-t002:** Age-Period-Cohort models for mortality due to trachea, bronchial and lung cancer in men aged ≥60 years. São Paulo-Brazil, 1980–2016.

Model	Resid	Resid	Deviance ^1^	*p* ^2^
	df	Dev		
**Osasco**				
Age	30	34.246		
Age−drift ^3^	29	33.273	0.9723	0.324
Age−Cohort	26	27.554	5.7197	0.126
Age−Period−Cohort	23	24.646	2.9079	0.406
Age−Period	26	31.494	−6.8482	0.076
Age−drift^4^	29	33.273	−1.7794	0.619
**Sorocaba**				
Age	30	50.262		
Age−drift ^3^	29	36.342	13.92	<0.001
Age−Cohort	26	26.611	9.7309	0.020
Age−Period−Cohort	23	25.494	1.1173	0.772
Age−Period	26	32.335	−6.8406	0.077
Age−drift^4^	29	36.342	−4.0077	0.26
**State of São Paulo**				
**Age**	30	707.23		
Age−drift ^3^	29	564.93	142.3	<0.001
Age−Cohort	20	164.08	400.85	<0.001
Age−Period−Cohort	15	40.22	123.86	<0.001
Age−Period	24	248	−207.77	<0.001
Age−drift ^4^	29	564.93	−316.93	<0.001

^1^ Deviance: Difference in degrees of freedom and deviance between the current model and the previous one, in the likelihood ratio test; ^2^
*p*: *p*-value of the likelihood ratio test; ^3^ Age-drift: Model in which the effect of age can be decomposed into period and cohort effects; ^4^ Age-drift: Model in which the longitudinal effect of age can be decomposed into effects of age and period.

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
