# Peer review of "Lung Cancer Mortality Trends in a Brazilian City with a Long History of Asbestos Consumption"

_ijerph, 2019, doi:10.3390/ijerph16142548_

Round 1

Reviewer 1 Report

Authors descrived lung cancer mortality terends in a Brazilian city, Osasco in São Paolo state, where possess long history of asbestos consumption, and compared it with whole São Paolo state as well as Sorocaba far from Osasco and does not have history of asbestos consumption.

The study design seemed to be well and manuscript is also well-written. Unfortunately, as authors also wrote in text as week points of this study, the Tabaco smoking backgrounds were not assumed. In addition, other causes for lung cancer such as air pollution monitoring were not extracted in this study. I assume Sorocaba is also the city of industrial city. If authors added other urban~agricultural city for comparison. Some findings between these differences of environmental situation may be extracted. However, it is not essential. I think the difference between two cities is asbestos consumption. Thus, this study is adapted to compare the asbestos-causing increase of lung cancer.

Authors can add these consideration in discussion part.

In addition, following small issues would be modified.

1) In my PDF file, I cannot read all figures. To small captions, too small lines. This should be changed

2) For figure 3 a, b and c, the center part of lines are started from “age area” in X-axis. This is something stragne

3) For future study, how authors hope to organize smoking status? Do you have any idea to resolve this? If you have, please write in discussion part.

Author Response

Dear editor,

            We thank you for the revisions to our manuscript submission. In the revised version the changes are highlighted in the text. We also uploaded the certificate of english editing by the American Journal Experts in the IJERPH website. 

            Please, find bellow the answers to each reviewer comments point by point.

Response to Reviewer 1 Comments

We thank you for your comments.  Please, find below the answers to each comment:

Point 1: The study design seemed to be well and manuscript is also well-written. Unfortunately, as authors also wrote in text as week points of this study, the Tabaco smoking backgrounds were not assumed. In addition, other causes for lung cancer such as air pollution monitoring were not extracted in this study.

Response 1: Thank you for your comment. The lack of information on the prevalence of tobacco smoking in Osasco is a weakness of the study. On the discussion section, we provided information on tobacco smoking available for São Paulo Capital as a proxy of  Osasco prevalence: “Since lung cancer is highly associated with tobacco smoking and air pollution, the lack of data on this specific risk factor for Osasco, Sorocaba and the State of São Paulo is a weakness of the study. Only data on smoking in São Paulo city (the State major city) is available from the 1970s, and it shows a reduction in smoking prevalence in men (54% in 1971, 32% in 1987, 25.5% in early 2001), and an decrease in women (20%, 32% and 19.8%, in 1971, 1987, and 2001, respectively) [43, 44, 45]; see Figure S3. In this study, we presumed that the trends for tobacco smoking would be expected to be the same across the locations and similar to those in São Paulo City.”

Based on your comments on air pollution monitoring, we included in the reviewed version the following sentence in discussion: “According to the Environmental Company of the State of São Paulo (CETESB), the level of particulate matter air pollution10 (<10 µm) decreased in the cities studied from 2000 to 2017 (Osasco=-46%; Sorocaba=-38%).”

Point 2: In my PDF file, I cannot read all figures. To small captions, too small lines. This should be changed.

Response 2: We increased the size of figures and captions. Thank you for the suggestion.

Point 3: For figure 3 a, b and c, the center part of lines are started from “age area” in X-axis. This is something stragne.

Response 3: Thank you for the comment. We improved the design of the graphic to turn it more clear.

Point 4: For future study, how authors hope to organize smoking status? Do you have any idea to resolve this? If you have, please write in discussion part.

Response 4: Unfortunately, there is no data on smoking in Osasco city available until now. We included in conclusions “In addition, the surveillance of the prevalence of major risk factors for lung cancer - such as tobacco smoking, asbestos exposure, ionizing radiation and air pollution- is important to increase the accuracy for more specific analysis of the fibre effect on lung cancer .”

Reviewer 2 Report

The biggest issue that may be obscuring the results found is lack of knowledge about smoking rates. It would also be useful to consider the percentage of the population of Osasco that might be engaged in asbestos related activities. Rates going down do, as noted, mirror what has happened elsewhere in men. Rates of smoking in women if known should be mentioned.Line 43 could use a refenrence though clearly true.

Author Response

Dear editor,

            We thank you for the revisions to our manuscript submission. In the revised version the changes are highlighted in the text. We also uploaded the certificate of english editing by the American Journal Experts in the IJERPH website. 

            Please, find bellow the answers to each reviewer comments point by point.

Response to Reviewer 2 Comments

We thank you for your comments.  Please, find below the answers to each comment:

Point 1: The biggest issue that may be obscuring the results found is lack of knowledge about smoking rates.

Response 1: Unfortunately, there is no available data on smoking specifically in Osasco city until now. To deal with this lack of information, we reported the prevalence of tobacco smoking in the biggest city of the State of São Paulo, which is conurbated with Osasco, as proxy of the prevalence in the areas studied. We then included the following sentences in the discussion section:  “Since lung cancer is highly associated with tobacco smoking and air pollution, the lack of data on this specific risk factor for Osasco, Sorocaba and the State of São Paulo is a weakness of the study. Only data on smoking in São Paulo city (the State major city) is available from the 1970s, and it shows a reduction in smoking prevalence in men (54% in 1971, 32% in 1987, 25.5% in early 2001), and an decrease in women (20%, 32% and 19.8%, in 1971, 1987, and 2001, respectively) [43, 44, 45]; see Figure S3. In this study, we presumed that the trends for tobacco smoking would be expected to be the same across the locations and similar to those in São Paulo City.”

Point 2: It would also be useful to consider the percentage of the population of Osasco that might be engaged in asbestos related activities.

Response 2: There is no official data on the prevalence of asbestos exposure in Osasco. Based on the estimation of population and worker’s registers, we included in discussion: “The estimated prevalence of asbestos exposure in the city of Osasco peaked at 1.27%, in 1988, among the economically active men”.

Point 3: Rates of smoking in women if known should be mentioned.

Response 3: We provided this important information in discussion, in the following sentences: “Only data on smoking in São Paulo city (the State major city) is available from the 1970s, and it shows a reduction in smoking prevalence in men (54% in 1971, 32% in 1987, 25.5% in early 2001), and an decrease in women (20%, 32% and 19.8%, in 1971, 1987, and 2001, respectively)”

Point 4: Line 43 could use a reference though clearly true.

Response 4: Thank you for the comment. We included a reference: (IARC, 2012)

“World Health Organization. International Agency for Research on Cancer (IARC). 2012. Arsenic, metals, fibres and dusts. Available online:  https://monographs.iarc.fr/iarc-monographs-on-the-evaluation-of-carcinogenic-risks-to-humans-19 (accessed on 18 june 2019)

Reviewer 3 Report

I have now read the manuscript submitted to IJREPH Journal by Gisele Aparecida Fernandes, Eduardo Algranti et al by the Dept of Epidemiology, School of Public Health, Univ of Sao Paulo, Brazil Entitled "Lung cancer Mortality trends in a Brazilian city with long history of asbestos consumption" My comments are as follows: 1) Brazil is currently in a notable focus of public health attention in view of its long time experience of asbestos product manufacturing and the studies of observations made in Sao Paolo State, the Sorocaba and Osasco regions (for reference populations) on lung cancer incidence and mortality. The Brazil publishing of scientific experience of monitoring lung cancer and airways cancer mortality is currently of particcular relevance in these times. It seems important to take such experiences into account in considering asbestos policies of nations and regions in the world public health scientific community. It deals with key issues of workers and public health with significant issues of public health to be monitored and supervised. 2) The group of authors in this case (Gisele Aparecide Fernandes et al) of the Sao Paolo University pwe it to the world  public health community to present high quality reports on their doings on the issues of exposure to inhaled fibres of Asbetos. 3) Authors are to declare intentions of include in such monitoring mesenchymal tumours of the pleura and peritonaeum (mesotheliomas) including steps to be taken for verification of pathological findings. This is necessary to retain crediblity in conducting proposed study. 4) Authors intentions on further monitoring of current exposure to inhaled asbestos fibres in index areas Sorocaba and Sao Paulo State and Osasco also need to be ensured. Assuming also responsibility for confoundig factors of tobacco smoking, inonizing radiation exposure  5)  A specialized statistical review of this submitted manuscript is also called for. In such review the statement offered by authors, on lines 113 - 114 in manuscript on assumption of "effects acting multiplicatively" with implications arising with linear effects on expected value of age, period and cohort effects deserve to be explained well to future readers for their proper understanding of statements made with reference to authors findings presented in figures two and three. This draft text passage needs to be well explained to readers.

Author Response

Dear editor,

            We thank you for the revisions to our manuscript submission. In the revised version the changes are highlighted in the text. We also uploaded the certificate of english editing by the American Journal Experts in the IJERPH website. 

            Please, find bellow the answers to each reviewer comments point by point.

Response to Reviewer 3 Comments

We thank you for your comments.  Please, find below the answers to each comment:

Point 1: Brazil is currently in a notable focus of public health attention in view of its long time experience of asbestos product manufacturing and the studies of observations made in Sao Paolo State, the Sorocaba and Osasco regions (for reference populations) on lung cancer incidence and mortality. The Brazil publishing of scientific experience of monitoring lung cancer and airways cancer mortality is currently of particular relevance in these times. It seems important to take such experiences into account in considering asbestos policies of nations and regions in the world public health scientific community. It deals with key issues of workers and public health with significant issues of public health to be monitored and supervised.

The group of authors in this case (Gisele Aparecide Fernandes et al) of the Sao Paolo University pwe it to the world  public health community to present high quality reports on their doings on the issues of exposure to inhaled fibres of Asbetos.

Response 1: We thank you for your comments.

Point 2: Authors are to declare intentions of include in such monitoring mesenchymal tumours of the pleura and peritonaeum (mesotheliomas) including steps to be taken for verification of pathological findings. This is necessary to retain crediblity in conducting proposed study.

Response 2: We included the following sentence in the discussion: “Another study limitation is the lack of comparison between lung cancer and mesothelioma mortality in the areas assessed. The number of mesothelioma cases is frequently used to estimate the number of lung cancer cases, the ratios varying according to the main fibre type consumed. In Brazil, most of the asbestos used was chrysotile. We were not able to assess age-period-cohort effects of mesothelioma mortality because of the reduced number of cases in the studied areas”.

Point 3: Authors intentions on further monitoring of current exposure to inhaled asbestos fibres in index areas Sorocaba and Sao Paulo State and Osasco also need to be ensured. Assuming also responsibility for confoundig factors of tobacco smoking, inonizing radiation exposure.

Response 3: We included in the conclusions: In addition, the surveillance of the prevalence of major risk factors for lung cancer - such as tobacco smoking, asbestos exposure, ionizing radiation and air pollution- is important to increase the accuracy for more specific analysis of the fibre effect on lung cancer .”

Point 4: A specialized statistical review of this submitted manuscript is also called for. In such review the statement offered by authors, on lines 113 - 114 in manuscript on assumption of "effects acting multiplicatively" with implications arising with linear effects on expected value of age, period and cohort effects deserve to be explained well to future readers for their proper understanding of statements made with reference to authors findings presented in figures two and three. This draft text passage needs to be well explained to readers.

Response 4: We included in the materials and methods: “The age-period-cohort model splits time into three axes: age, period, and cohort. These models were built for verification, in particular in our study, of the period effect. Such an effect often results from external factors affecting the age groups over a specific period of time. In the cohort effect, similar health outcomes are observed in groups of individuals who were born in similar years. In the effect of age, the influence of time on health outcomes is observed.”

Round 2

Reviewer 1 Report

Authors modified their manuscript according to the reviewer's comments.

Reviewer 2 Report

appreciate updates as given.